# The effect of long-range linkage disequilibrium on allele-frequency dynamics under stabilizing selection

Sherif Negm[1], Carl Veller[2,3]*

1 Department of Human Genetics, University of Chicago, Chicago, Illinois, United States of America,
2 Department of Ecology & Evolution, University of Chicago, Chicago, Illinois, United States of America,
3 National Institute for Theory and Mathematics in Biology, Northwestern University and University of Chicago, Chicago, Illinois, United States of America

* cveller@uchicago.edu

## Abstract

Stabilizing selection on a polygenic trait reduces the trait's genetic variance by (i) generating correlations (linkage disequilibria) between opposite-effect alleles throughout the genome, and (ii) selecting against rare alleles at loci that affect the trait, eroding heterozygosity at these loci. Here, we show that the linkage disequilibria, which stabilizing selection generates on a rapid timescale, slow down the subsequent allele-frequency dynamics at individual loci, which proceed on a much longer timescale. Exploiting this separation of timescales, we obtain expressions for the expected per-generation change in minor-allele frequency at individual loci, as functions of the effect sizes at these loci, the strength of selection on the trait, its variance and heritability, and the linkage relations among loci. Using whole-genome simulations, we show that our expressions predict allele-frequency dynamics under stabilizing selection more accurately than the formulae that have previously been used for this purpose. Our results have implications for understanding the genetic architecture of complex traits.

## Author summary

Stabilizing selection—selection for optimal trait values—is likely pervasive across humans and other species. Its phenotypic effect is to reduce trait variance, and it achieves this genetically by favoring compensating combinations of trait-increasing and trait-decreasing variants throughout the genome, generating correlations between them, and by selecting against rare variants at individual loci. We show that the correlations generated by stabilizing selection slow the rate at which it purges rare variants. We characterize this effect mathematically,

**Data availability statement:** The SLiM script used to generate the figures in this paper is available at https://doi.org/10.5061/dryad.np5hqc089.

**Funding:** This work was supported by a Branco Weiss Fellowship (to CV). The funders had no role in study design, data collection and analysis, decision to publish, or preparation of the manuscript.

**Competing interests:** The authors have declared that no competing interests exist.

and show via simulations that the expressions we derive for the frequency dynamics at individual loci are accurate. Our results make possible more precise detection and quantification of stabilizing selection in genomic data.

## 1 Introduction

To understand the genetic architecture of polygenic traits, we need to connect population genetic models with genomic data such as those from genome-wide association studies (GWASs). For many traits, a particularly plausible model is stabilizing selection, which penalizes deviations from an optimal trait value. Theoretical argument and empirical evidence indicate that many complex traits are under stabilizing selection [1]. For example, in humans, Sanjak et al. [2] used lifetime reproductive success as a proxy for fitness and estimated significant nonlinear selection gradients consistent with stabilizing selection for a large fraction of the traits they analyzed. Furthermore, a number of studies have demonstrated for many human traits that the joint distribution of allele frequencies and effect sizes (inferred from GWAS) is consistent with stabilizing selection [3–5] (see also [6]).

The macroscopic consequence of stabilizing selection on a complex trait is to reduce the trait's genetic variance over time. This is achieved in two ways. First, by selecting for compensating combinations of trait-increasing and trait-decreasing alleles, stabilizing selection rapidly generates negative correlations—linkage disequilibria (LD)—between alleles with the same directional effect on the trait [7] (this has come to be known as the 'Bulmer effect'). Second, stabilizing selection generates weak selection against the rarer allele at each polymorphic locus affecting a complex trait, slowly eroding heterozygosity at these loci [8,9].

This second variance-reducing effect—which mimics fitness underdominance at trait-affecting loci—can be understood intuitively in the following way (see also Section 3.1 below). Consider a focal polymorphic locus affecting the trait, and suppose that the common allele at the locus is trait-increasing, so that the mean effect of the locus is to increase the trait. For the population's average trait value to be at the optimum, the mean effect of the rest of the genome must compensate for the mean effect at the focal locus, and must therefore be trait-decreasing. But this leaves the rare allele at the focal locus—which is also trait-decreasing—maladapted to the rest of the genome: on average, it resides in individuals with trait values further from the optimum than the common allele at the locus does.

The speed of the resulting allele-frequency dynamics depends, as the intuition above suggests, on the mean phenotypes experienced by the trait-increasing and trait-decreasing alleles at the focal locus. These mean phenotypes are determined not only by the individual effects of the alleles themselves in concert with the mean effect of the rest of the genome, but also by the effects of alleles with which the alleles at the focal locus are associated via LD. Because of the Bulmer effect, alleles at the focal locus will tend to be associated with opposite-effect alleles elsewhere in the genome, partially masking the individual effects of the alleles at the focal locus.

This brings the mean phenotypes experienced by the alleles at the focal locus closer to the optimum, weakening selection between them and therefore slowing down their frequency dynamics.

Here, we quantify the slowdown in allele-frequency dynamics at individual loci caused by the Bulmer effect. Exploiting a separation of the timescales over which stabilizing selection generates LD (rapidly) and changes allele frequencies (slowly; S1 Fig), we obtain simple expressions for the expected change in an allele's frequency across a single generation under stabilizing selection, as a function of its individual effect, the strength of stabilizing selection on the trait, the trait's variance and heritability, and the linkage relations among loci in the genome. As we show, the expressions that we derive predict allele-frequency change in simulations more accurately than other expressions that have commonly been used.

There is a deep theoretical literature on the population genetics of stabilizing selection, and some of the conclusions that we reach echo results from earlier work (see especially [10–14] and syntheses in [15–17]). For example, assuming a normal distribution of allelic effects on a trait under stabilizing selection, Lande [10] and Turelli & Barton [13] found that the linkage relations among loci do not affect the trait's additive genetic variance at equilibrium. This can be understood as reflecting a balance between two consequences of the stronger negative LD that accumulates with tighter linkage under stabilizing selection. First, that the stronger negative LD, as one component of the trait's genetic variance, directly reduces the genetic variance by a greater amount. Second, that the stronger negative LD more severely slows down the allele-frequency dynamics at individual loci, allowing the rarer allele at each locus to be maintained at a higher mutation–selection balance in expectation, thus indirectly increasing the other component of the trait's additive genetic variance: the variance contributed by polymorphism at individual loci (the 'genic variance').

However, as in the example above, previous treatments have usually focused on the consequences of the Bulmer effect for the equilibrium values of aggregate quantities such as the genetic and phenotypic variance of the trait under selection (but see [18]). For many applications though, it is important to characterize the per-locus allele-frequency dynamics themselves. For example, the joint distribution of allele frequencies and effect sizes is an important summary of a trait's genetic architecture and, under stabilizing selection, is determined by the allele-frequency dynamics at individual loci [3–5]. Additionally, because the allele-frequency dynamics under stabilizing selection are very slow, analyses of their equilibria implicitly assume long-term constancy of the strength of stabilizing selection, which may be inappropriate for many traits (e.g., [19]; see Discussion). It might therefore be useful to understand the impact of the Bulmer effect on allele-frequency dynamics outside equilibrium scenarios. Finally, the theoretical literature on stabilizing selection is often highly technical. While this has allowed very general results to be obtained, it has also perhaps limited the absorption of these results into the empirical literature, and into human genetics in particular. It is therefore an important challenge to derive simple, intuitive results for population genetic dynamics under stabilizing selection that are both accurate and portable into existing empirical frameworks.

## 2 Model

We consider an additive polygenic trait under stabilizing selection. Genetic variation in the trait is contributed by $L$ autosomal polymorphic loci, $l = 1, 2, \ldots, L$. At each locus $l$, there is a trait-increasing allele, with frequency $p_l$ and haploid effect $+\alpha_l/2$, and a trait-decreasing allele, with frequency $1 - p_l$ and haploid effect $-\alpha_l/2$ (so that the difference in the phenotypes of a homozygote for the trait-increasing allele and a homozygote for the trait-decreasing allele is, all else equal, $2\alpha_l$). Since we are interested in characterizing the effect of selection on allele-frequency dynamics at these loci, we ignore mutation (although we later discuss the role of mutation and the implications of our results for the rate of turnover of the loci underlying genetic variation in a trait—see Discussion).

An individual's trait value $Y$ is given by

$$Y = G + E,$$

where

$$G = \sum_{l=1}^{L} \alpha_l g_l$$

is the individual's additive genetic value for the trait, with their genotype $g_l$ coded as −1, 0, or 1 if they carry 0, 1, or 2 trait-increasing alleles at locus $l$ respectively. $E$ is an environmental disturbance that we assume to be independent of $G$ and to have mean zero.

We denote the phenotypic variance by $V_P = \text{Var}(Y)$, the additive genetic variance by $V_G = \text{Var}(G)$, and the environmental variance by $V_E = \text{Var}(E)$. The trait's heritability is $h^2 = V_G/V_P$, and its genic variance—the additive genetic variance ignoring the contribution of linkage disequilibrium (LD) among loci—is $V_g = \sum_{l=1}^{L} 2p_l(1 - p_l)\alpha_l^2$.

The trait is under stabilizing selection around an optimal value that we arbitrarily code as 0: the relative fitness of an individual with trait value $Y$ is specified by the Gaussian fitness function

$$\phi(Y) = e^{-\frac{Y^2}{2V_S}}, \tag{1}$$

where $V_S$ modulates the strength of stabilizing selection on the trait (with smaller values of $V_S$ corresponding to stronger selection). It will sometimes be useful to approximate this Gaussian fitness function by a first-order Taylor approximation around the optimal value of zero,

$$\phi(Y) \approx 1 - \frac{Y^2}{2V_S}. \tag{2}$$

We assume that mating is random and that the trait is not affected by any forms of selection other than the stabilizing selection specified above.

Under these assumptions, upon the onset of selection, the mean trait value in the population rapidly converges to the optimal value of zero (e.g., [20,21]). We are interested in the allele-frequency dynamics at causal loci after this directional phase of selection, once stabilizing selection has commenced, and we therefore assume that the population starts with a mean trait value equal to zero—with the allele frequencies $p_l$ such that this is the case.

## 3 Results

Since we are interested in the allele-frequency dynamics at individual loci, it will be useful to consider the marginal fitnesses of the two alleles at a given locus, that is, the average fitness of an individual carrying a randomly chosen copy of the one allele versus the other. From these marginal fitnesses, we can calculate an 'effective selection coefficient' $s_{\text{eff}}$ for one of the alleles—say, the minor allele. The expected change in frequency of this allele across a single generation is then $\mathbb{E}[\Delta p] = p(1 - p)s_{\text{eff}}$, where $p$ is its frequency in the earlier generation [22].

We begin with a simple calculation of these marginal fitnesses under the model described above, ignoring phenotypic variation from the environment and from loci other than the focal locus, and also ignoring LD between the focal locus and other loci. We will then add in these two ingredients in turn.

### 3.1 A simple one-locus calculation

Consider a focal locus segregating for a trait-increasing allele $A$, at frequency $p$ and with effect size $+\alpha/2$, and a trait-decreasing allele $a$, at frequency $1 - p$ and with effect size $-\alpha/2$ (we have omitted the locus subscript since we are considering a single locus in isolation here).

A haploid instance of this locus contains allele $A$ with probability $p$ and allele $a$ with probability $1 - p$. The average genetic value at a haploid instance of the locus is therefore

$$p \times (+\alpha/2) + (1 - p) \times (-\alpha/2) = (2p - 1)\alpha/2.$$

Under stabilizing selection around an optimal trait value of zero, the average genome-wide genetic value must be zero. Therefore, considering the average genetic value of the focal haploid locus above, the genetic value of the rest of the genome (including the homologous instance of the locus) must be $-(2p - 1)\alpha/2 = (1 - 2p)\alpha/2$.

Now suppose that the allele at the focal haploid locus is $A$. If we ignore any correlation between the allelic state at the locus and the genetic value of the rest of the genome—i.e., if we ignore LD—the average genetic value of the individual carrying this $A$ allele is

$$\underbrace{+\alpha/2}_{\substack{\text{contribution of} \\ \text{the allele itself}}} + \underbrace{(1 - 2p)\alpha/2}_{\substack{\text{average contribution} \\ \text{of rest of genome}}} = (1 - p)\alpha.$$

$$(3)$$

If the allele at the focal haploid locus is instead $a$, the average genetic value of the individual is

$$\underbrace{-\alpha/2}_{\substack{\text{contribution of} \\ \text{the allele itself}}} + \underbrace{(1 - 2p)\alpha/2}_{\substack{\text{average contribution} \\ \text{of rest of genome}}} = -p\alpha.$$

$$(4)$$

Thus, the average phenotypes inhabited by the $A$ and $a$ alleles differ in sign. If $p < 1/2$, that is, if $A$ is the minor allele at the locus, the average phenotype inhabited by an $A$ allele will be further from zero than the average phenotype inhabited by an $a$ allele (and vice versa if $p > 1/2$). This explains why the minor allele is selected against under stabilizing selection: for the average value of the phenotype to be at its optimum, the rest of the genome must adapt to the more common allele at the locus, leaving the rarer allele maladapted to the rest of the genome.

(As an interesting aside, we may use the same method to calculate the average phenotypes experienced by the three possible diploid genotypes at the focal locus, and therefore to rank the fitnesses of the three genotypes. The mean phenotypic effect of the diploid locus is twice the mean haploid effect, i.e., $(2p - 1)\alpha$, and so the mean phenotypic effect of all other loci is $(1 - 2p)\alpha$. Therefore, again ignoring systematic signed LD among causal loci, the mean phenotype of the $aa$ genotype is $-\alpha + (1 - 2p)\alpha = -2p\alpha$, that of the $Aa$ genotype is $0 + (1 - 2p)\alpha = (1 - 2p)\alpha$, and that of the $AA$ genotype is $+\alpha + (1 - 2p)\alpha = 2(1 - p)\alpha$. Assume that $A$ is the minor allele at the locus, i.e., that $p < 1/2$. Then, if $p < 1/4$, $|-2p\alpha| < |(1 - 2p)\alpha| < |2(1 - p)\alpha|$, i.e., the fitness ranking of the genotypes is $w(aa) > w(Aa) > w(AA)$. If, instead, $1/4 < p < 1/2$, then $|(1 - 2p)\alpha| < |-2p\alpha| < |2(1 - p)\alpha|$, i.e., the fitness ranking is $w(Aa) > w(aa) > w(AA)$. In neither case is the heterozygous genotype the least fit, and in fact it is the fittest genotype when the minor-allele frequency is greater than 1/4 [9]. Therefore, although the frequency dynamics at the locus (Eq. 5) resemble those expected under classical underdominance at the locus, the fitness values of the genotypes at the locus do not.)

If we further ignore the phenotypic variation contributed by the environment and from the rest of the genome, then we can substitute the average phenotypic values above into our fitness function to calculate the marginal fitnesses of $A$ and $a$. Employing the quadratic approximation in Eq. (2), the mean fitness of the bearer of a randomly chosen $A$ allele is

$$w_A = 1 - \frac{(1 - p)^2\alpha^2}{2V_S},$$

while the mean fitness of the bearer of a randomly chosen $a$ allele is

$$w_a = 1 - \frac{p^2 \alpha^2}{2V_S}.$$

Since the mean fitness in the population $\bar{w}$ is close to 1, the effective selection coefficient of $A$ is

$$s_A^{\text{eff}} = \frac{w_A - w_a}{\bar{w}} \approx w_A - w_a = \frac{\left[p^2 - (1-p)^2\right]\alpha^2}{2V_S^2} = \left(p - \frac{1}{2}\right)\frac{\alpha^2}{V_S},$$

which is positive if $p > 1/2$ (when $A$ is the common allele at the locus) and negative if $p < 1/2$ (when $A$ is the rare allele at the locus).

The expected change in frequency of $A$ across a single generation is then

$$\mathbb{E}\left[\Delta p\right] = p(1-p)s_A^{\text{eff}} = p(1-p)\left(p - \frac{1}{2}\right)\frac{\alpha^2}{V_S}, \tag{5}$$

which is a commonly used formula for allele-frequency change under stabilizing selection (e.g., [3,8,9]).

However, this formula overpredicts the rate of allele-frequency change in simulations (Fig 1). This is for two reasons. First, phenotypic variation from the environment and from other loci obscures the signal that selection sees of each allele's average phenotype. Second, the alleles come into positive LD with opposite-effect alleles elsewhere in the genome, bringing the average phenotypes of their bearers closer to zero and thus weakening selection at the locus. We consider these two effects in turn.

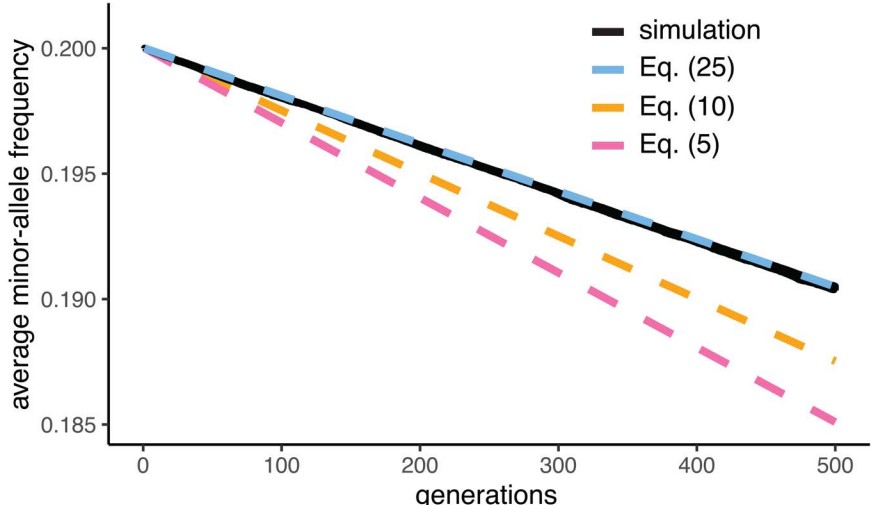

**Fig 1. Simulated versus predicted average minor-allele frequencies across the polymorphic loci affecting a trait under stabilizing selection, in the case of full symmetry across loci.** In the simulations, there are $L = 1,000$ polymorphic loci, all unlinked and with equal effect sizes ($\alpha = 1$) and starting minor-allele frequencies ($\min\{p_l, 1 - p_l\} = 0.2$). The loci are initially in linkage equilibrium, and the strength of selection $V_S$ is chosen such that $V_S/V_G = 5$ initially. Further simulation details can be found in the Methods. Eq. (25) (dashed blue line), which takes into account background phenotypic variance and the Bulmer effect, is seen to be a better prediction of the observed trajectory of average minor-allele frequency (solid black line) than either Eq. (5) (dashed pink line), which ignores both background phenotypic variance and the Bulmer effect, and Eq. (10) (dashed yellow line), which ignores the Bulmer effect. The simulation trajectory is averaged over 1,000 replicate trials.

## 3.2 Background phenotypic variance

We assume that the trait value is normally distributed with mean zero and variance $V_P$. Still ignoring LD for now, the distribution of trait values experienced by $A$ alleles is then normal with mean $(1-p)\alpha$ (from Eq. 3) and variance $\sim V_P$ (because, for a highly polygenic trait, the genetic variance at the individual haploid locus is small compared to the overall phenotypic variance). Similarly, the distribution of trait values experienced by $a$ alleles is normal with mean $-p\alpha$ (from Eq. 4) and variance $\sim V_P$.

In general, if the trait distribution among the bearers of a given allele is normal with mean $\mu$ and variance $V_P$, then the average fitness of the bearer of a randomly chosen copy of the allele is

$$w(\bar{y} = \mu) = e^{-\frac{\mu^2}{2(V_S+V_P)}},$$

(6)

as shown in refs. [3,20] and S1 Text Section S1. The effect of taking into account background variance in the trait is simply to dilute the strength of selection at the focal locus by a factor $1 + V_P/V_S$, which is greater if the phenotypic variance is large relative to the width of the selection function.

Therefore, in our case, the mean relative fitness of the bearer of a randomly chosen $A$ allele is

$$w_A = w(\bar{y} = (1-p)\alpha) = e^{-\frac{(1-p)^2\alpha^2}{2(V_S+V_P)}} \approx 1 - \frac{(1-p)^2\alpha^2}{2(V_S+V_P)},$$

(7)

while the mean relative fitness of the bearer of a randomly chosen $a$ allele is

$$w_a = w(\bar{y} = -p\alpha) = e^{-\frac{p^2\alpha^2}{2(V_S+V_P)}} \approx 1 - \frac{p^2\alpha^2}{2(V_S+V_P)},$$

(8)

where the approximations hold because $\alpha^2 \ll V_S + V_P$ for a polygenic trait. Since the overall mean fitness is close to 1, the effective selection coefficient of $A$ is

$$s_A^{\text{eff}} = \frac{w_A - w_a}{\bar{w}} \approx w_A - w_a = \frac{p^2\alpha^2}{2(V_S+V_P)} - \frac{(1-p)^2\alpha^2}{2(V_S+V_P)} = \left(p - \frac{1}{2}\right)\frac{\alpha^2}{V_S+V_P},$$

(9)

and so the expected change in frequency of $A$ across a single generation is

$$\mathbb{E}\left[\Delta p\right] = p(1-p)s_A^{\text{eff}} = p(1-p)\left(p - \frac{1}{2}\right)\frac{\alpha^2}{V_S+V_P}.$$

(10)

This prediction of allele-frequency change is smaller than the prediction of Eq. (5) by a factor $V_S/(V_S + V_P)$, but, while more accurate than Eq. (5), Eq. (10) still overpredicts the rate of allele-frequency change in simulations (Fig 1). As we show next, this is because the mean phenotypes of bearers of $A$ and $a$ are in fact closer to zero than predicted by Eqs. (3) and (4), because $A$ and $a$ have each come into LD with opposite-effect alleles elsewhere in the genome.

## 3.3 The Bulmer effect

As we have discussed, stabilizing selection reduces a trait's genetic variance in two ways. The first is that, by favoring compensating combinations of trait-increasing and trait-decreasing alleles, stabilizing selection generates negative LD

between alleles with the same directional effect on the trait. This leads to a rapid decline in the trait's genetic variance, to a quasi-equilibrium value that reflects a balance between the generation of LD by selection and its destruction by recombination (S1 Fig). The second is that stabilizing selection induces selection against rare alleles affecting the trait, eroding heterozygosity on average at their loci. For a highly polygenic trait, selection against rare alleles is weak, and so their average frequency decline is slow.

In the limit of high polygenicity and a large population size, allele frequencies do not change at all under selection, and so the reduction in a trait's genetic variance due to stabilizing selection is entirely due to LD. In this limit, and under some other simplifying assumptions including that all loci are unlinked, Bulmer [7] calculated the equilibrium reduction in the genetic variance of a trait under stabilizing selection, as a function of the trait's initial genetic variance in the absence of LD (its genic variance, $V_g$), the strength of stabilizing selection on the trait ($V_S$), and the contribution of the environment to trait variance ($V_E$). Bulmer [23] extended this calculation to allow for variable linkage among loci (see also [15]).

An allele's LD with causal alleles elsewhere in the genome will affect the mean trait value experienced by the allele, and therefore also its frequency dynamics under selection. To calculate how the LD generated by stabilizing selection affects allele-frequency dynamics at individual loci, we first determine how LD of any kind affects the mean trait value experienced by an allele. Thereafter, we calculate the total degree of LD generated by stabilizing selection at equilibrium, and the expected apportionment of this overall amount of LD to specific pairs of loci. Exploiting the difference in the timescales over which LD is generated and rare alleles decline in frequency under stabilizing selection (S1 Fig), we then substitute the equilibrium degree of LD into our calculation for how LD affects the mean trait value experienced by an allele to find the expected effect of LD on allele-frequency dynamics under stabilizing selection. The close agreement between our analytical predictions and our simulations validates this separation-of-timescales approach for polygenic traits.

**3.3.1 The effect of LD on the mean trait value experienced by an allele.** Recall that there are $L$ loci underlying genetic variation in the trait, and that the trait-increasing allele at locus $l$ is at frequency $p_l$ and increases the value of the trait by $\alpha_l/2$, while the trait-decreasing allele is at frequency $1 - p_l$ and decreases the trait value by $\alpha_l/2$. We denote the coefficient of LD between the trait-increasing alleles at loci $l$ and $l'$ by $D_{ll'}$. Since the population mean value of the trait is, by assumption, at its optimum 0,

$$\sum_{l=1}^{L} \left[ p_l \times \left( +\frac{\alpha_l}{2} \right) + (1 - p_l) \times \left( -\frac{\alpha_l}{2} \right) \right] = 0 \quad \Rightarrow \quad \sum_{l=1}^{L} (2p_l - 1)\alpha_l = 0. \tag{11}$$

Define the random variable $I_l$ to take the value 1 if the allele at locus $l$ in a randomly chosen haploid/gametic genome is trait-increasing, and 0 if the allele is instead trait-decreasing. From Bayes' theorem, these indicator variables are related to the coefficients of LD via

$$\text{Prob}(I_{l'} = 1 \mid I_l = 1) = p_{l'} + D_{ll'}/p_l. \tag{12}$$

Consider a copy of the trait-increasing allele at $l$, and suppose, without loss of generality, that it is maternally inherited. Because mating is random, the mean genetic value of the paternally inherited genome is that of a randomly chosen haploid genome, i.e., 0. So the mean trait value experienced by the allele is the mean genetic value of the maternally inherited genome in which it lies. (Similarly, if it is paternally inherited, the mean trait value it experiences is the mean genetic value of the paternally inherited genome.) Therefore, the average trait value experienced by a trait-increasing allele at $l$ is

$$\mathbb{E}[Y \mid I_l = 1] = \underbrace{\left(+\frac{\alpha_l}{2}\right)}_{\substack{\text{contribution of}\\\text{trait-incr. allele}\\\text{at focal locus } l}} + \underbrace{\sum_{l' \neq l}\left[\text{Prob}(I_{l'} = 1 \mid I_l = 1)\left(+\frac{\alpha_{l'}}{2}\right) + [1 - \text{Prob}(I_{l'} = 1 \mid I_l = 1)]\left(-\frac{\alpha_{l'}}{2}\right)\right]}_{\substack{\text{average contribution of rest of (haploid) genome,}\\\text{conditional on having trait-increasing allele at } l}}$$

$$= \frac{1}{2}\left(\alpha_l + \sum_{l' \neq l}\left[2\,\text{Prob}(I_{l'} = 1 \mid I_l = 1) - 1\right]\alpha_{l'}\right)$$

$$= \frac{1}{2}\left(\alpha_l + \sum_{l' \neq l}\left[2p_{l'} - 1 + \frac{2D_{ll'}}{p_l}\right]\alpha_{l'}\right)$$

$$= \frac{1}{2}\left(2(1-p_l)\alpha_l + \sum_{l'=1}^{L}\left[2p_{l'} - 1\right]\alpha_{l'} \!\!\!\!\diagup\!\!\!\! + \sum_{l' \neq l}\frac{2D_{ll'}\alpha_{l'}}{p_l}\right)$$

$$= (1 - p_l)\alpha_l + \sum_{l' \neq l}\frac{D_{ll'}\alpha_{l'}}{p_l}$$

$$= (1 - p_l)\left(\alpha_l + \frac{1}{p_l(1-p_l)}\sum_{l' \neq l}D_{ll'}\alpha_{l'}\right). \tag{13}$$

Sensibly, positive LD with trait-increasing alleles elsewhere in the genome ($D_{ll'} > 0$) will tend to increase the trait value experienced by the trait-increasing allele at $l$, while negative LD with other trait-increasing alleles ($D_{ll'} < 0$) will tend to decrease the trait value it experiences. With no LD ($D_{ll'} = 0$), $\mathbb{E}[Y \mid I_l] = (1 - p_l)\alpha_l$, recovering Eq. (3) above.

By a similar calculation, the average trait value experienced by a randomly chosen trait-decreasing allele at $l$ is

$$\mathbb{E}[Y \mid I_l = 0] = -p_l\left(\alpha_l + \frac{1}{p_l(1-p_l)}\sum_{l' \neq l}D_{ll'}\alpha_{l'}\right). \tag{14}$$

**3.3.2 Symmetric loci.** The calculations above for the mean trait values experienced by the two alleles at a locus $l$ hold for any nature and source of the LD terms $D_{ll'}$ (see Discussion). Since our particular interest is in stabilizing selection as a source of LD, we now calculate these terms in expectation under stabilizing selection. We begin with the simplest possible case, where all loci are unlinked and have the same minor-allele frequency $p$ and effect size $\alpha$. In this case, the mean phenotype experienced by a randomly chosen trait-increasing allele at locus $l$ is, from Eq. (13),

$$\mathbb{E}[Y \mid I_l = 1] = (1-p)\alpha\left(1 + \frac{1}{p(1-p)}\sum_{l' \neq l}D_{ll'}\right), \tag{15}$$

while the mean phenotype experienced by a randomly chosen trait-decreasing allele at $l$ is, from Eq. (14),

$$\mathbb{E}[Y \mid I_l = 0] = -p\alpha\left(1 + \frac{1}{p(1-p)}\sum_{l' \neq l}D_{ll'}\right); \tag{16}$$

that is, LD between the focal locus and other causal loci in the genome causes the mean phenotypes experienced by both alleles at the focal locus to be multiplied by a factor $1 + \frac{1}{p(1-p)}\sum_{l' \neq l}D_{ll'}$, as if the effect size at the locus were not $\alpha$ but instead $\alpha\left(1 + \frac{1}{p(1-p)}\sum_{l' \neq l}D_{ll'}\right)$.

Under the same conditions of symmetry across loci, and in the limit of high polygenicity, the equilibrium reduction $d$ in the trait's genetic variance due to the Bulmer effect satisfies

$$\frac{d}{V_g} = \frac{3 + X - \sqrt{1 + 6X + X^2}}{4},$$

(17)

where $X = (V_S + V_E)/V_g$ ([7]; S1 Text Section S3.4 in [24]). (Note that, for neatness, we define $d$ to be the *reduction* in the genetic variance, so that it is a positive number; this is in contrast to Bulmer's $d$, which is the net change in the genetic variance and therefore negative.) When $V_S \gg V_g$, $d/V_g \approx V_g/(V_S + V_E)$ [25].

While Eq. (17) gives the total reduction in the trait's genetic variance explicitly in terms of the model parameters $V_S$, $V_E$, and $V_g$ (which do not themselves depend on $d$), in practice we will usually not have access to measurements of a trait's genic variance $V_g$ (since we cannot characterize all of the loci that contribute variance to the trait). Fortunately, the equilibrium value of $d/V_g$ can also be expressed implicitly in terms of the heritability of the trait, $h^2$, and its variance, $V_P$ (which both depend on $d$), as well as $V_S$:

$$\frac{d}{V_g} = \frac{h^2}{1 + h^2 + V_S/V_P},$$

(18)

where $h^2$ and $V_P$ are at their quasi-equilibrium values (see S1 Text Section S3.4.3 in [24]). Eq. (18) obviates the need to estimate $V_g$.

The reduction $d$ of the trait's genetic variance is entirely due to LD among the polymorphic loci that underlie genetic variation in the trait:

$$-d = 2\alpha^2 \sum_{l=1}^{L} \sum_{\substack{l'=1 \\ l' \neq l}}^{L} D_{ll'} \quad \Rightarrow \quad \sum_{l=1}^{L} \sum_{\substack{l'=1 \\ l' \neq l}}^{L} D_{ll'} = -\frac{d}{2\alpha^2}.$$

(19)

Under the assumption of symmetry across loci in their effects and minor-allele frequencies, this total sum of LD is apportioned equally, in expectation, across the $L(L-1)/2$ pairs of loci: for each pair of loci $l$ and $l'$,

$$\mathbb{E}[D_{ll'}] = -\frac{d}{2\alpha^2 L(L-1)}.$$

(20)

(Note that the summations in Eq. (19) count each pair of loci twice.) So, for a given causal locus $l$, the sum of its LD coefficients with the other $L-1$ causal loci is, in expectation,

$$\mathbb{E}\left[ \sum_{\substack{l'=1 \\ l' \neq l}}^{L} D_{ll'} \right] = -\frac{d}{2\alpha^2 L}.$$

(21)

Substituting Eq. (21) into Eqs. (15) and (16), we find that the mean trait value of a randomly chosen trait-increasing allele at locus $l$ is

$$\mathbb{E}[Y \mid l_l = 1] = (1-p)\alpha \left( 1 - \frac{d}{2p(1-p)\alpha^2 L} \right)$$

$$= (1-p)\alpha \left( 1 - \frac{d}{V_g} \right)$$

(22)

and, similarly, that the mean trait value of a randomly chosen trait-decreasing allele at locus $l$ is

$$\mathbb{E}[Y \mid I_l = 0] = -p\alpha \left(1 - \frac{d}{V_g}\right).$$

(23)

Comparing Eqs. (22) and (23) with Eqs. (3) and (4), we see that the mean phenotypes of the two alleles at the locus are as they would be in the simple scenario considered before with no LD, but with the effect sizes of the alleles attenuated by a factor $1 - d/V_g$ (with $d/V_g$ specified by Eq. 17). That is, the LD generated by stabilizing selection can be incorporated into the classical formulae for allele-frequency change at the locus (Eqs. 5 and 10), which ignore LD, by defining an 'effective' effect size,

$$\alpha_{\text{eff}} = \alpha \left(1 - \frac{d}{V_g}\right).$$

(24)

Across a single generation, the expected change in frequency of the trait-increasing allele at the focal locus is then

$$\mathbb{E}[\Delta p] = p(1-p)\left(p - \frac{1}{2}\right)\frac{\alpha_{\text{eff}}^2}{V_S + V_P}$$

$$= p(1-p)\left(p - \frac{1}{2}\right)\frac{\alpha^2 \left(1 - \frac{d}{V_g}\right)^2}{V_S + V_P}.$$

(25)

(Note that the phenotypic variances experienced by the trait-increasing and trait-decreasing alleles at $l$ are still equal, and close to $V_P$ owing to the trait's high polygenicity.)

A reviewer has suggested an alternative approach to deriving Eqs. (13) and (14), and thereby (25), which we briefly outline here. The trait value $Y$ is a linear regression on the genotype $g_l$ at any given locus $l$, and the slope of this regression, $\text{Cov}(Y, g_l)/\text{Var}(g_l)$, has denominator $2p_l(1 - p_l)$ and a numerator that is a sum of effect-size-weighted LD terms $D_{ll'}\alpha_{l'}$ ($2D_{ll'}\alpha_{l'}$ being the covariance between the genotype at $l$ and the contribution of $l'$ to the trait). Combining this observation with Eqs. (3) and (4), one can obtain Eqs. (13) and (14) as fitted values of the regression for $g_l = +1$ and $g_l = -1$ respectively. A similar approach is taken by Bulmer [15, Ch. 10], who obtains an expression (his Eq. 10.12) that can be translated to Eq. (25).

In simulations of stabilizing selection under the conditions of symmetry across loci assumed here, Eq. (25) is seen to be a better predictor of the allele-frequency dynamics at causal loci than either Eq. (5) or Eq. (10) (Fig 1).

Relaxing the assumptions that effect sizes and minor-allele frequencies are the same across loci, we show in S1 Text Section S3 that Eq. (25) is still a close approximation to the frequency trajectory at an individual locus when all loci are unlinked (S2 Fig).

The parameters in Fig 1 involve relatively strong stabilizing selection ($V_S/V_g \approx 5$). These parameters were chosen primarily because they generate strong LD, leading to larger disparities between the predictions that do and do not take this LD into account, and to 'stress test' the way in which Eq. (25) incorporates this LD. However, they are also consistent with the median quadratic selection gradient among those compiled by Kingsolver et al. [26] that are consistent with stabilizing selection ([27]; see also discussion in [17, Ch. 28]). Nonetheless, the prediction of Eq. (25) remains accurate in simulations of weak stabilizing selection as well, $V_S/V_g \approx 25$ (S3 Fig), of the order of that estimated to act on human height [2,28].

Our expression for the expected change in allele frequencies at loci affecting the trait, Eq. (25), furthermore allows us to predict the change in the genic variance $V_g$ over time, via the relation

$$\mathbb{E}\left[\Delta V_g\right] = \sum_l 2\alpha_l^2 \left((1-2p_l)\mathbb{E}[\Delta p_l] - \left(\mathbb{E}\left[\Delta p_l\right]\right)^2 - \frac{p_l(1-p_l)}{2N_e}\right) \tag{26}$$

derived in S1 Text Section S2. In S4 Fig, we show that Eq. (25), when substituted into Eq. (26), provides a much better prediction of the trajectory of the genic variance observed in simulations than Eqs. (5) and (10).

### 3.3.3 Incorporating linkage.
The calculations above assume full symmetry across loci, and, in particular, that every locus is on a separate chromosome. This symmetry allowed us to apportion the total amount of LD generated by stabilizing selection evenly, in expectation, among the locus pairs.

We now consider the case of variable linkage relations among loci, with some locus pairs lying on the same chromosome and some lying on different chromosomes. Let the recombination fraction between loci $l$ and $l'$ be $r_{ll'}$. For now, we maintain the assumption of constant effect sizes and minor-allele frequencies across loci. Variable linkage relations among loci affect the total amount of LD generated by stabilizing selection: In the limit of high polygenicity, the equilibrium reduction $d$ in the trait's genetic variance due to the Bulmer effect satisfies

$$\frac{d}{V_g} = \frac{1 - \bar{r}_h\left(\sqrt{1 + 2\left(1 + \frac{1}{\bar{r}_h}\right)X + X^2} - (1 + X)\right)}{1 + 2\bar{r}_h}, \tag{27}$$

where $X = (V_S + V_E)/V_g$ as before, and $\bar{r}_h = L(L-1)/\left(\sum_l\sum_{l'\neq l}\frac{1}{r_{ll'}}\right)$ is the harmonic mean recombination rate across loci ([23]; S1 Text Section S3.4 in [24]). $\bar{r}_h$ can be estimated from various kinds of data, including linkage maps, cytological data, and sequence data, similarly to the arithmetic mean recombination rate $\bar{r} = \sum_l\sum_{l'\neq l} r_{ll'}/L(L-1)$ [29]. When $V_S \gg V_g$, $d/V_g \approx \frac{1}{2\bar{r}_h} \cdot \frac{V_g}{V_S + V_E}$.

As in the case of no linkage, in practice we do not need to estimate $X = (V_S + V_E)/V_g$, and in particular $V_g$, to estimate the equilibrium value of $d/V_g$. Instead, it can be expressed implicitly in terms of $V_P$ and $h^2$ (which depend on $d$), as well as $V_S$ and $\bar{r}_h$:

$$\frac{d}{V_g} = \frac{h^2}{2\bar{r}_h(1 + V_S/V_P) + h^2}, \tag{28}$$

where, again, $h^2$ and $V_P$ are taken to be at their quasi-equilibrium values (see S1 Text Section S3.4.3 in [24]).

As before, the reduction $d$ of the trait's genetic variance is entirely due to LD among loci, with $\sum_l\sum_{l'\neq l} D_{ll'} = -d/(2\alpha^2)$. However, now, this overall LD is not apportioned equally among locus pairs in expectation. Instead, the expected LD between $l$ and $l'$ is proportional to the inverse of the recombination fraction between the loci $1/r_{ll'}$ [23] (this relationship is true as long as $r_{ll'}$ is not too small), so that

$$\mathbb{E}[D_{ll'}] = -\frac{d}{2\alpha^2 L(L-1)} \cdot \frac{\bar{r}_h}{r_{ll'}}. \tag{29}$$

For a given locus $l$, let $\bar{r}_h^{(l)} = (L-1)/\sum_{l'\neq l}\left(1/r_{ll'}\right)$ be the harmonic mean recombination fraction between $l$ and the $L-1$ other causal loci. Then, summing Eq. (29) across these $L-1$ other loci, we find

$$\mathbb{E}\left[\sum_{\substack{l'=1 \\ l'\neq l}}^{L} D_{ll'}\right] = -\frac{d\bar{r}_h}{2\alpha^2 L(L-1)}\sum_{l'\neq l}\frac{1}{r_{ll'}} = -\frac{d}{2\alpha^2 L} \cdot \frac{\bar{r}_h}{\bar{r}_h^{(l)}}. \tag{30}$$

Because of our assumption of equal effect sizes and minor-allele frequencies across loci, Eqs. (15) and (16) still specify the mean trait values experienced by a randomly selected trait-increasing and trait-decreasing allele at $l$ respectively. Substituting Eq. (30) into (15) and (16), we find

$$\mathbb{E}[Y \mid I_l = 1] = (1-p)\alpha \left( 1 - \frac{d}{2p(1-p)\alpha^2 L} \cdot \frac{\bar{r}_h}{\bar{r}_h^{(l)}} \right)$$

$$= (1-p)\alpha \left( 1 - \frac{d}{V_g} \cdot \frac{\bar{r}_h}{\bar{r}_h^{(l)}} \right) \tag{31}$$

and, similarly,

$$\mathbb{E}[Y \mid I_l = 0] = -p\alpha \left( 1 - \frac{d}{V_g} \cdot \frac{\bar{r}_h}{\bar{r}_h^{(l)}} \right). \tag{32}$$

Therefore, comparing the equations above with those that ignore LD (Eqs. 3 and 4), we can again think of the impact of the LD generated by stabilizing selection on the mean phenotypes experienced by the alleles at a particular locus in terms of an attenuation of the 'effective' effect size at the locus, with this 'effective' effect size dependent on the recombination relations of the locus to other causal loci, according to

$$\alpha_{\text{eff}}^{(l)} = \alpha \left( 1 - \frac{d}{V_g} \cdot \frac{\bar{r}_h}{\bar{r}_h^{(l)}} \right). \tag{33}$$

Intuitively, alleles at loci that have tighter recombination relations with other loci ($\bar{r}_h/\bar{r}_h^{(l)} > 1$) will develop stronger average LD with the opposite-effect alleles at these other loci, and will therefore have their individual effects more greatly masked—and so their frequency dynamics more severely slowed—by the Bulmer effect than alleles at loci with looser recombination relations with other loci ($\bar{r}_h/\bar{r}_h^{(l)} < 1$).

The expected change in frequency of the minor allele at $l$ across a single generation is

$$\mathbb{E}\left[\Delta p_l\right] = p_l(1-p_l) \left( p_l - \frac{1}{2} \right) \frac{\left( \alpha_{\text{eff}}^{(l)} \right)^2}{V_S + V_P}$$

$$= p_l(1-p_l) \left( p_l - \frac{1}{2} \right) \frac{\alpha^2 \left( 1 - \frac{d}{V_g} \cdot \frac{\bar{r}_h}{\bar{r}_h^{(l)}} \right)^2}{V_S + V_P}. \tag{34}$$

Taking the average of Eq. (34) across loci provides a prediction of the average change in the minor alleles' frequencies. This prediction agrees well with simulations that make use of realistic linkage maps (Fig 2). Particularly in the case of a low-recombination species (*Drosophila melanogaster*; $\bar{r}_h \approx 0.07$), in which the LD generated by the Bulmer effect is especially strong, Eq. (34) offers a much improved prediction of the allele-frequency dynamics over Eqs. (5) and (10) (Fig 2B).

We show in S1 Text Section S3 that, if we relax the assumptions that effect sizes and minor-allele frequencies are the same across loci, Eq. (34) remains a close approximation to the frequency trajectory at locus $l$, and we verify this by simulation under both the *D. melanogaster* linkage map (S5 Fig) and the human map (S6 Fig).

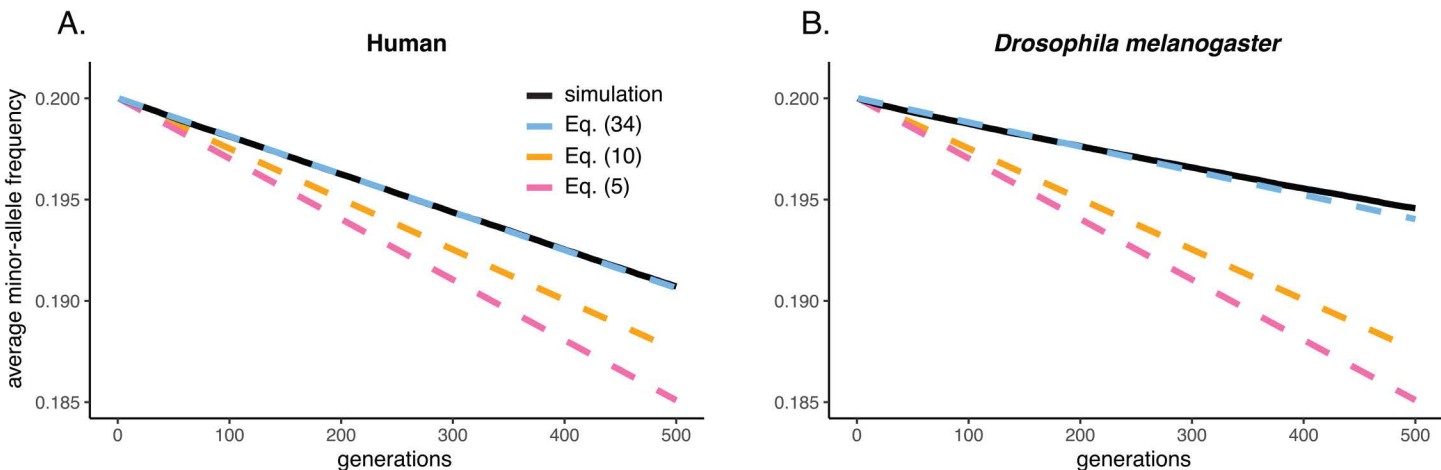

**Fig 2. Average change in the minor-allele frequency at polymorphic loci affecting a trait under stabilizing selection, when these loci are distributed across the human linkage map (A) and the linkage map of *Drosophila melanogaster* (B).** Eq. (34) is seen to predict simulated trajectories of the average minor-allele frequency better than Eqs. (5) and (10), especially for *D. melanogaster*, a low-recombination species. Other than the variable linkage relations among loci, simulation details are identical to Fig 1.

Although Fig 2 shows the correspondence of our predictions of allele-frequency changes and those observed in simulations under relatively strong selection ($V_S/V_g \approx 5$), our predictions remain accurate under weaker selection as well ($V_S/V_g \approx 25$), for both the human and *D. melanogaster* linkage maps (S3 Fig).

As in the case with no linkage, substituting Eq. (34) into Eq. (26) provides a much better prediction of the observed change in the genic variance in simulations with the human and *D. melanogaster* linkage maps than substituting Eqs. (5) and (10) into (26) does (S4 Fig).

We can also check the prediction of Eq. (34) for loci with lower and higher locus-specific harmonic mean recombination rates $\bar{r}_h^{(l)}$. Collecting loci in bins of gradually higher values of $\bar{r}_h^{(l)}$ across the *D. melanogaster* linkage map (which, unlike in high-recombination species like humans, shows a sizeable range of $\bar{r}_h^{(l)}$ values), Fig 3 plots the average minor-allele frequency in each bin after 250 generations of selection against the average value of $\bar{r}_h^{(l)}$ within each bin. While the prediction of Eq. (34) is a substantial improvement over predictions that do not take into account LD (and therefore also ignore variable linkage relations among loci), Eq. (34) underpredicts the degree of allele-frequency change for the lowest-recombination bins.

The reason has to do with the separation-of-timescales assumption underlying Eq. (34), and in particular the assumption that the quasi-equilibrium degree of LD (Eq. 29) is attained instantaneously for every pair of loci at the onset of selection. In reality, the expected degree of LD between each pair of loci builds up over time, approaching its equilibrium value at a rate that depends on the recombination fraction between the pair of loci involved: very rapidly for loosely linked locus pairs, but more slowly for tightly linked locus pairs. Therefore, for the most tightly linked locus pairs, the Bulmer effect is, for an appreciable number of generations after the onset of selection, weaker than assumed by Eq. (34), and so the allele-frequency dynamics at these loci are faster in these early generations than predicted by our calculations. The result is that, after a given number of generations, allele frequencies have changed more than predicted by Eq. (34), with the disparity greater for more tightly-linked loci—the pattern observed in Fig 3.

We can, in fact, for every pair of loci, calculate the expected LD in each generation after the onset of selection, and use these values to obtain a more accurate—albeit more complicated—prediction for the overall allele-frequency change at each locus after a given number of generations. Initially, the degree of LD between each pair of loci $l$ and $l'$ is zero: $D_{ll'}^{(0)} = 0$. Within each generation $t$, selection generates an increment to the degree of disequilibrium between $l$ and $l'$ equal

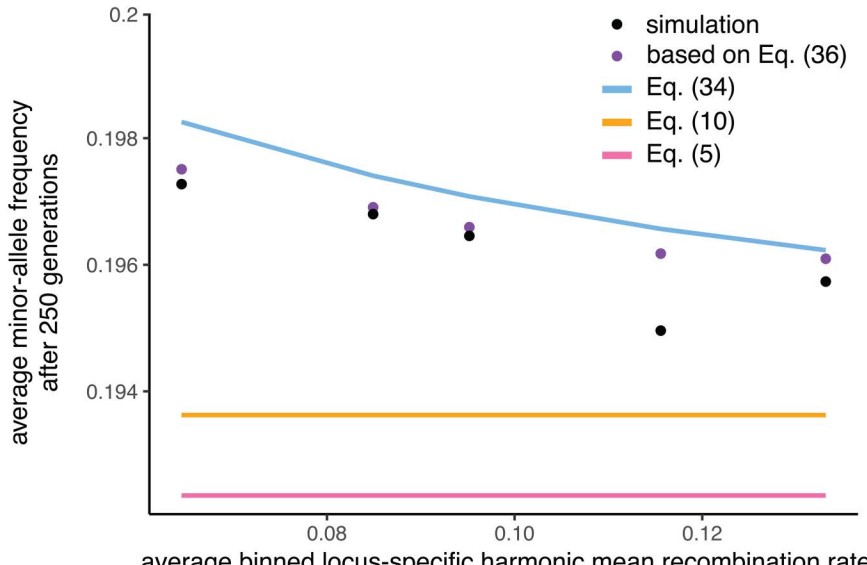

**Fig 3. Minor-allele frequencies after 250 generations of selection, for loci with different average recombination rates with the rest of the genome.** The simulations here are the same as for Fig 2B, with loci distributed along the *D. melanogaster* linkage map. For the simulated minor-allele frequencies, loci have been binned according to their locus-specific harmonic mean recombination fractions with other loci, $\bar{r}_h^{(l)}$. While the equilibrium-based prediction (Eq. 34) that takes into account the Bulmer effect (blue line) is a substantial improvement over predictions that do not take into account the Bulmer effect (Eqs. 5 and 10; pink and yellow lines), it underpredicts the degree of allele-frequency change, especially for loci with low values of $\bar{r}_h^{(l)}$, since for these values, the approach to equilibrium values of LD is slow. A prediction based on the full sequence of non-equilibrium values of LD (Eq. 36; details in Methods) performs better than the equilibrium-based prediction. Note that the fourth and fifth bins contain only 6 and 2 loci respectively, potentially explaining the discrepancy between the simulated minor-allele frequencies and the predictions based on Eq. (36). Simulation points are averages across 500 replicate trials.

to $2K$ (we will determine the value of $K$ shortly). This amount $2K$ is the same in expectation for all pairs of loci, because we have assumed that effect sizes are constant, and is divided approximately equally in expectation between cis-LD (LD among pairs of alleles inherited from the same parent) and trans-LD (LD among alleles inherited from different parents), which selection does not distinguish [15,23]. In transmission to the next generation $t + 1$, recombination between loci $l$ and $l'$ generates an amount $r_{ll'}K$ of new cis-LD from the trans-LD that built up in generation $t$, while an amount $(1 - r_{ll'})K$ of the cis-LD that built up in generation $t$ is preserved in transmission to the generation $t + 1$, along with an amount $(1 - r_{ll'})D_{ll'}^{(t)}$ of the cis-LD that was already present at the beginning of generation $t$ (before selection acted). Therefore,

$$\mathbb{E}\left[D_{ll'}^{(t+1)} \mid D_{ll'}^{(t)}\right] = r_{ll'}K + (1 - r_{ll'})K + (1 - r_{ll'})D_{ll'}^{(t)} = K + (1 - r_{ll'})D_{ll'}^{(t)}. \tag{35}$$

Starting with zero LD ($D_{ll'}^{(0)} = 0$), Eq. (35) defines a sequence of expected LD values

$$D_{ll'}^{(1)} = K, \quad D_{ll'}^{(2)} = \left(1 + (1 - r_{ll'})\right)K, \quad D_{ll'}^{(3)} = \left(1 + (1 - r_{ll'}) + (1 - r_{ll'})^2\right)K, \quad \dots, \tag{36}$$

which converges at rate $r_{ll'}$ to an asymptotic value of $K/r_{ll'}$ [15,23]. This is the equilibrium degree of LD between $l$ and $l'$; comparison with Eq. (29) therefore reveals that $K = -d\bar{r}_h/2\alpha^2 L(L-1)$.

For each polymorphic locus $l$ affecting the trait, Eq. (36) thus specifies the expected LD with every other polymorphic locus $l'$ affecting the trait, and so, for each generation $t$ after the onset of selection, we can substitute these into Eqs. (13) and (14) to calculate the mean trait value experienced by the trait-increasing and trait-decreasing alleles at $l$; from these,

we can calculate the expected change in frequency of the alleles at locus $l$ from generation $t$ to generation $t + 1$ (see Methods). Carrying out this calculation, we find that it predicts the cumulative allele-frequency change observed across generations in our simulations more successfully than Eq. (34), especially for loci with especially tight average linkage relations with the other loci in the genome (Fig 3).

The expressions in Eq. (36) for the individual pairwise linkage disequilibria each generation after the onset of selection further allow us to predict the change in the contribution $d$ of these linkage disequilibria to the genetic variance $V_G = V_g - d$. Together with Eq. (26), which relates changes in allele frequencies to changes in the genic variance $V_g$, this allows us to predict the expected change in the genetic variance $V_G$ over time (S1 Text Section S2). S4 Fig shows that employing Eqs. (25) and (34) in this procedure leads to accurate predictions of the trajectory of the genetic variance, both when all loci are unlinked and when they lie along the linkage maps of humans and *D. melanogaster*.

## 4  Discussion

Understanding the processes that govern the genetic architecture of complex traits will require the interpretation of genomic data in terms of population genetic models. The richest and finest-scale data come from genome-wide association studies, which offer estimates of allelic effect sizes at thousands of sites throughout the genome [30]. Because of the per-site nature of GWAS data, its interpretation in terms of population genetic models will require a detailed understanding of allele-frequency dynamics under these models.

Here, we have provided simple calculations that predict allele-frequency dynamics under stabilizing selection—a common mode of selection on complex traits [1,2]—more accurately than the formulae that have previously been used for this purpose. To do so, we have incorporated into these formulae the linkage disequilibrium that stabilizing selection rapidly generates between opposite-effect alleles throughout the genome [7,23].

The accuracy of our calculations in predicting simulated allele-frequency trajectories under stabilizing selection (Fig 1–3) suggests that they may make possible more precise quantitative interpretation of GWAS and other genomic data in terms of population genetic models of stabilizing selection. Below, we discuss some of the implications of our results for the interpretation of such data.

### 4.1  Genetic architecture of complex traits

Several studies have shown that, for many human traits, the joint distribution of allele frequencies and effect sizes (as inferred from GWAS) is consistent with the allele-frequency dynamics expected under stabilizing selection [3–5]. In demonstrating this consistency, these studies made use of Eq. (5) as a description of the allele-frequency dynamics. As we have shown, Eq. (5) overpredicts the rate of allele-frequency change under stabilizing selection, because it ignores the systematically signed LD generated by stabilizing selection (and background phenotypic variance of the trait under selection). Importantly, however, the equations that we have derived for allele-frequency change that do take into account the LD generated by stabilizing selection are qualitatively of the same form as those that ignore this LD, with an 'effective' effect size of each allele substituted for the allele's true effect size (Eqs. 24, 33). Therefore, the results of [3–5] are not qualitatively affected by the more accurate predictions of allele-frequency change under stabilizing selection that we have derived. However, their estimates for the strength of stabilizing selection on human traits must be revised upwards, since for a given strength of stabilizing selection on a trait, the slowdown of the allele-frequency dynamics induced by the Bulmer effect will result in higher minor-allele frequencies, on average, across loci.

### 4.2  Allelic turnover and the portability of polygenic scores across populations

Once a GWAS has been carried out in a particular sample, it is common to use the effect-size estimates obtained from the GWAS to generate polygenic scores (PGSs)—sums of an individuals' genotypes across trait-associated loci, with each locus weighted in the sum by its estimated effect size—and to measure the accuracy of these PGSs as predictors of trait

values both within the original GWAS sample and in other samples. Such exercises have revealed that PGSs are often much worse predictors of trait values in samples that are more distantly related to the original GWAS sample (e.g., [31,32]).

Several explanations have been offered for this 'portability' problem, including differences between populations in (i) the effect sizes of causal alleles, (ii) the frequencies of causal alleles, (iii) the degree to which genotyped SNPs 'tag' causal alleles (via close-range LD), and (iv) the environmental and genetic effects with which variation at genotyped SNPs is confounded (reviewed in [33]). Of interest here is the second explanation, that PGSs can suffer reduced portability because of differences between populations in allele frequencies at causal loci [34–37]. In some cases, an ancestral polymorphism might have been retained in a sample drawn from one population but lost in a sample drawn from another; the locus would then contribute trait variation in the one sample but not in the other.

Although turnover of the loci underlying variation in a trait is possible by neutral drift alone, the process is accelerated—and the portability of PGS prediction across populations more rapidly degenerates—if the alleles at these loci are under selection [36–40]. Yair & Coop [36] note that stabilizing selection on the trait itself will have such an effect, because stabilizing selection speeds up allele-frequency dynamics relative to neutral drift. Yair & Coop further recognized that the allele-frequency dynamics under stabilizing selection, and consequently the rate of allelic turnover between populations, would be slowed by the Bulmer effect (especially for ensembles of tightly linked-loci; see their Figure S1). However, for simplicity, in quantifying the effect of stabilizing selection on the reduction of trait variance contributed by ancestral polymorphisms, and the consequent degeneration of PGS prediction accuracy across descendent populations, they ignored the contribution of LD generated by the Bulmer effect.

Since, as we have shown, the impact of the Bulmer effect on allele-frequency dynamics under stabilizing selection can be captured by defining 'effective' effect sizes of causal alleles, it can be incorporated into the calculations of Yair & Coop in a simple way. For example, their Eq. (2.3) calculates the fraction of trait variance in a generation-$t$ descendent population explained by polymorphisms of effect size $\alpha$ that were present in the ancestral population: $\exp(-F_{ST}[1 + S/4])$, where $F_{ST} = t/2N$ is the value of $F_{ST}$ between the ancestral and descendant populations at neutral loci and $S = 2N\alpha^2/V_S$. This calculation can be modified to take into account the Bulmer effect simply by replacing $\alpha$ with the 'effective' effect size $\alpha_{\text{eff}}$, as defined in Eqs. (24) and (33). The same is true of the analogous calculations developed by Yair & Coop based on a diffusion approximation.

### 4.3  Assumptions, and notions of equilibrium under stabilizing selection

Much previous work on the genetic dynamics of stabilizing selection, including that which considers the impact of the Bulmer effect, has been carried out under the assumption of equilibrium. The broadest definition of equilibrium under stabilizing selection is that the distribution of trait values is stationary, with the trait mean at its optimum and the trait variance held constant by a balance between selection, which reduces it, and mutation, which replenishes it. Although the mean will rapidly attain its optimal value if it is initially displaced from it [20,21], the time it takes for the variance to subsequently equilibrate is much longer, because the allele-frequency dynamics under stabilizing selection are very slow.

Therefore, studies that assume (or characterize) a constant distribution of trait values—and in particular a constant variance—under stabilizing selection implicitly assume long-term constancy of the strength of selection, the population's demography, etc. The strictness of this assumption has allowed powerful results to be obtained, for example, in showing that the genetic variance at equilibrium under stabilizing selection is sometimes independent of the genetic map [10,13] and in characterizing the equilibrium allele-frequency spectrum as a function of effect size [3,5]. But the assumption that demography and the strength of selection have been constant on extremely long timescales might be problematic for many of the traits and populations to which such analyses might be relevant.

In contrast, our analysis of allele-frequency dynamics under stabilizing selection depends on relatively few assumptions. We have assumed that the trait mean is at its optimum, which, as noted above, it will very soon be if it starts away

from the optimum (given sufficient genetic variance). We have also assumed that the negative LD that stabilizing selection generates is at its equilibrium value—this too occurs on a rapid timescale [23] (S1 Fig). Thus, our results hold for a wide range of non-equilibrium scenarios. For example, if the strength of selection changes, our expressions for allele-frequency change will become accurate as soon as the (now stronger) LD has re-equilibrated, while analyses based on trait equilibrium must wait much longer for the variance to re-equilibrate.

In our calculations of allele-frequency dynamics at individual loci, we have ignored the influence of mutation at these loci. This is because our interest is in the genetic dynamics of complex traits, which are now known to have extremely large mutational targets (e.g., $> 10^7$ for human height, estimated by Simons et al. [4]), only a small fraction of which will be sufficiently polymorphic at any given time to contribute meaningfully to the trait's genetic variation (e.g., $\sim 10^4$ for human height [41,42]). For such traits, therefore, most new trait-affecting mutations occur at loci that are not currently polymorphic; consequently, the frequency dynamics at polymorphic loci are largely unaffected by mutation, and therefore closely follow the expressions we have derived. One way in which mutation could affect our results, however, is if new mutations tend to have a particular directional effect on the trait value, such as to decrease it on average, since this would lead to persistent one-sided departures of the trait mean from its optimum and thus directional selection on allele frequencies [43].

An interesting question concerns the robustness of our results to changes in demography, such as population size and structure. The equilibrium amount of LD under stabilizing selection, and the rate at which it is attained, do not depend on the population's size or structure (being a product of selection). It is true that, in smaller populations, the mean value of the trait tends to deviate further from the optimum, but these deviations are negligible unless the population size is very small relative to the strength of selection (the trait mean explores a stationary distribution with variance $V_S/2N_e$ [20]). Therefore, the two key assumptions underlying our calculations are largely unaffected by demography. As with the strength of selection, since populations' demography will seldom be constant over the long timescales required for trait equilibrium to be reached under stabilizing selection, the robustness of our results to changes in demography is a major advantage.

Nonetheless, demography can affect the rate of allele-frequency dynamics at individual loci. This is because the rate of the frequency dynamics is mediated by the genic variance $V_g$, via its contribution both to $V_P$ in the denominators of Eqs. (25) and (34) and to the slowdown factor $1 - d/V_g$ in their numerators, and because $V_g$ itself can depend on population demography in complex ways (e.g., [44]).

### 4.4 Other processes that generate long-range signed LD

While stabilizing selection itself generates systematically signed long-range LD between alleles that affect the trait under selection (see [45] for recent evidence of this effect in humans), with these LDs impacting the frequency dynamics of the alleles in the way we have described, other processes can also generate systematically signed LD, and these LDs too will influence allele-frequency dynamics under stabilizing selection.

Most important among these other processes is assortative mating, which is known to be common for traits in humans [46,47] and other animals [48]. Assortative mating for a heritable trait generates positive LD between alleles with like effects on the trait [49–51]. If the trait is also under stabilizing selection, this positive LD will tend to counteract the negative LD among like-effect alleles generated by stabilizing selection, speeding up the rate at which the minor allele declines in frequency relative to the case where there is no assortment on the trait. Whether the frequency dynamics are ultimately faster or slower than expected in the absence of any LD depends on whether the negative LD generated by stabilizing selection is outweighed by, or outweighs, the positive LD generated by assortative mating. Our general calculation of the mean trait values experienced by the two alleles at a locus, as a function of their effects and their degrees of LD with other causal loci (Eqs. 13 and 14), allows this balance to be quantified.

As with the LD generated by stabilizing selection itself, the effect of LD generated by assortative mating on allele-frequency dynamics under stabilizing selection can be captured by 'effective' effect sizes of alleles at polymorphic

loci affecting the trait. In contrast to the case of stabilizing selection, under assortative mating, the equilibrium degree of LD expected between a pair of loci does not depend on the recombination fraction between the loci [50]. It can be calculated as a simple function of the correlation of the trait value among mates, the trait's heritability, and the effect sizes of the alleles involved [24,50,52].

Suppose, for simplicity, that effect sizes $\alpha$ and minor-allele frequencies $p$ are equal across loci. If the heritability of the trait is $h^2$ and the phenotypic correlation among mates is $\rho$, then the expected LD between the trait-increasing alleles at loci $l$ and $l'$ due to assortative mating is $D_{ll'}^{\text{assort}} = \frac{h^2\rho}{1-h^2\rho} \cdot \frac{p(1-p)}{2(L-1)}$ (see SI Section S3.1.1 in [24]). Therefore, for a given locus $l$, the expected sum of the LDs between its trait-increasing allele and the trait-increasing alleles at other loci, due to assortative mating, is

$$\mathbb{E}\left[\sum_{\substack{l'=1 \\ l' \neq l}}^{L} D_{ll'}^{\text{assort}}\right] = \frac{h^2\rho}{1-h^2\rho} \cdot \frac{p(1-p)}{2} = \frac{h^2\rho}{1-h^2\rho} \cdot \frac{V_g}{4L\alpha^2}.$$

(37)

If, when the trait is under assortative mating and stabilizing selection jointly, the LD between a pair of loci is simply the sum of the LDs expected under the two processes separately ($D_{ll'} = D_{ll'}^{\text{assort}} + D_{ll'}^{\text{stab}}$), then, assuming all loci to be unlinked, we combine Eqs. (37) and (21) to find

$$\mathbb{E}\left[\sum_{\substack{l'=1 \\ l' \neq l}}^{L} D_{ll'}\right] = \mathbb{E}\left[\sum_{\substack{l'=1 \\ l' \neq l}}^{L} D_{ll'}^{\text{assort}}\right] + \mathbb{E}\left[\sum_{\substack{l'=1 \\ l' \neq l}}^{L} D_{ll'}^{\text{stab}}\right]$$

$$= \frac{h^2\rho}{1-h^2\rho} \cdot \frac{V_g}{4L\alpha^2} - \frac{d}{2L\alpha^2}$$

$$= \frac{V_g}{4L\alpha^2}\left(\frac{h^2\rho}{1-h^2\rho} - \frac{2d}{V_g}\right),$$

(38)

where $d/V_g$ is given by Eq. (17). Substitution into Eqs. (13) and (14) reveals the effective effect size of each allele in this case to be

$$\alpha_{\text{eff}} = \alpha\left(1 + \frac{h^2\rho}{2(1-h^2\rho)} - \frac{d}{V_g}\right).$$

(39)

If we allow for variable linkage relations among loci, a similar calculation combining Eqs. (37) and (30) yields

$$\mathbb{E}\left[\sum_{\substack{l'=1 \\ l' \neq l}}^{L} D_{ll'}\right] = \frac{V_g}{4L\alpha^2}\left(\frac{h^2\rho}{1-h^2\rho} - \frac{2d}{V_g} \cdot \frac{\bar{r}_h}{\bar{r}_h^{(l)}}\right),$$

(40)

and an effective effect size

$$\alpha_{\text{eff}}^{(l)} = \alpha\left(1 + \frac{h^2\rho}{2(1-h^2\rho)} - \frac{d}{V_g} \cdot \frac{\bar{r}_h}{\bar{r}_h^{(l)}}\right),$$

(41)

where $d/V_g$ is now given by Eq. (27). On average across loci, therefore, $\mathbb{E}_l\left[\mathbb{E}\left[\sum_{l' \neq l} D_{ll'}\right]\right] = \frac{V_g}{4L\alpha^2}\left(\frac{h^2\rho}{1-h^2\rho} - \frac{2d}{V_g}\right)$, as in the case with no linkage (though with a larger value of $d$ owing to linkage).

Therefore, if $2d/V_g > h^2\rho/(1 - h^2\rho)$, the impact of stabilizing selection will outweigh that of assortative mating, the effective effect sizes of alleles will be attenuated from their true effects, and allele-frequency dynamics under stabilizing selection will be slowed. In contrast, if $2d/V_g < h^2\rho/(1 - h^2\rho)$, the impact of assortative mating outweighs that of stabilizing selection, the effective effect sizes of alleles are amplified, and allele-frequency dynamics under stabilizing selection are accelerated.

As an example, human height is under strong assortative mating ($\rho \sim 0.25$ [53]) and moderately strong stabilizing selection ($V_S/V_P \sim 20$–30 [2,28]). Assuming its heritability to be $h^2 = 0.8$, and making use of the value $\bar{r}_h = 0.464$ we have calculated and the estimate $V_S/V_P \sim 30$ from Sanjak et al. [2], we find that $2d/V_g \sim 0.054$, while $h^2\rho/(1 - h^2\rho) = 0.25$. Therefore, the effective effects of loci that affect height are amplified, on average, because of the relatively strong assortative mating; their frequency dynamics under stabilizing selection are therefore expected to be accelerated, in spite of the Bulmer effect.

Note that patterns of trait-based migration can lead to long-range LD similar in nature to that under assortative mating [24].

## 4.5 Pleiotropy

Our calculations have ignored the possibility that the alleles underlying variation in the focal trait also affect other traits which might be under stabilizing selection (or, following the discussion above, might be subject to other processes that generate long-range signed LD). To see how such pleiotropy could affect our conclusions, consider a pair of alleles at distinct loci that both increase the focal trait. All else equal, we would expect stabilizing selection on the focal trait to generate negative LD between these two alleles, to a degree that we can calculate in expectation (e.g., Eqs. 20 and 29). However, suppose that there is another trait under stabilizing selection, and the one allele increases this trait but the other allele decreases it. All else equal, we would expect stabilizing selection on this other trait to generate positive LD between the alleles. Clearly, the sign and magnitude of the LD between the alleles will depend on the size of their effects on the two traits, as well as the relative strengths of selection on the two traits.

While ultimately the degree and nature of pleiotropy is an empirical question, we can suggest some qualitative predictions for how the Bulmer effect will influence the frequency dynamics of alleles that affect multiple traits under stabilizing selection. Following Simons et al. [3,4], we focus on a scenario where we have allelic effect-size estimates for a single trait, but where these alleles also affect other, unmeasured traits.

Consider first the 'isotropic' case, where the effects of an allele on the various traits are independent. This is the case considered by Simons et al. [3,4], who note that it will always apply in a suitably reconfigured coordinate system for the traits in question. Consider two alleles that increase the focal trait. In the absence of pleiotropy, we would expect these alleles to come into negative LD because of selection on the focal trait. And because, under isotropic pleiotropy, the signs and magnitudes of the alleles' effects on other traits are independent of their effects on the focal trait, the LD between the alleles is unchanged, *in expectation*, by their effects on these other traits. The realized value of LD between this particular pair of alleles will change because of pleiotropy, but across such locus pairs, there is no tendency for pleiotropy to systematically increase or decrease the degree of LD from the value expected under selection on the focal trait alone.

It is, however, perhaps more likely that alleles' effects on traits will be correlated. That is, if a pair of alleles both increase trait A, and we learn that one of the alleles decreases trait B, then the other allele probably also decreases trait B (since, for example, the effects of the alleles on traits A and B might in part be mediated through shared pathways in a genetic network). In this case, alleles with like effects on the focal trait we have measured are disproportionately likely to have like effects on other, unmeasured traits. In this case, the magnitude of the LD between a pair of alleles would, in expectation, exceed the value expected based on their effects on the focal trait alone, as calculated, for example, in Eqs. (20) and (29). Therefore, in this perhaps more realistic scenario than the isotropic model, the slowdown in allele-frequency dynamics induced by the Bulmer effect would be more severe than we have calculated.

## 5 Conclusion

Stabilizing selection is increasingly recognized as a key driver of the genetic architecture of complex traits [1,3–5,36]. The calculations that we have developed here make possible more precise predictions of its effects on genetic architectures and, conversely, more precise inference of the parameters of stabilizing selection from genomic data.

As we have noted, there is a rich theoretical literature on the population genetics of stabilizing selection, in which the Bulmer effect has been extensively treated (e.g., [7,10–15,23]). We see our work as complementary to this literature in its focus on the consequences of the Bulmer effect for single-locus allele-frequency dynamics rather than aggregate quantities such as genetic and phenotypic variance. In this, and in its relative simplicity, our work offers a potential bridge between this prior theoretical literature and practical application to the growing body of population genomic data.

## Methods

All simulations were run in SLiM 4.0 [54]. The model organism is diploid, sexual, and undergoes random mating each generation. In all setups, the population size is $N = 10,000$ and there are initially $L = 1,000$ autosomal polymorphic loci, either all unlinked or placed along a linkage map as described below. Each individual's trait value is specified by $Y = \sum_{l=1}^{L} \alpha_l g_l$, where $g_l$ is the number of trait-increasing alleles that the individual carries at locus $l$ minus 1, and $\alpha_l$ is the effect size at the locus, as described in the Model section of the Main Text. For simplicity, in our simulations there is no environmental component of trait variation; i.e., the trait is fully heritable. The fitness of an individual with trait value $Y$ is given by $\phi(Y) = \exp(-Y^2/2V_S)$. In each simulation, $V_S$ was chosen so that $V_S/V_P = 5$, where $V_P$ is the initial variance of the trait (equal to the initial genetic variance $V_G$, since the trait is by assumption fully heritable). Code is available at https://doi.org/10.5061/dryad.np5hqc089 [55].

**Methods for Fig 1.** The $L$ loci are all unlinked with respect to each other (specified by recombination rate 1/2 between 'adjacent' loci in SLiM), and the effect size at each locus $l$ is $\alpha_l = 1$. At 500 of these loci, the initial frequency of the trait-increasing allele is $p_l = 0.2$, while at the other 500 loci, it is $p_l = 0.8$. The designation of the loci at which the trait-increasing allele is common versus rare is arbitrary, since all loci are unlinked with respect to each other. Trait-increasing and trait-decreasing alleles are initially assigned randomly to haploid genomes, independently across loci, so that there is no LD among loci in expectation. The initial additive genetic variance, $V_G$, is therefore equal in expectation to the initial genic variance, $V_g = \sum_{l=1}^{L} 2p_l(1-p_l)\alpha_l^2 = 320$, and so the inverse strength of selection is $V_S = 5 \times 320 = 1,600$.

The simulation then proceeds for 500 generations, with fitness in each generation specified as described above. In each generation, we measure the frequency of the (initially) minor allele at each locus, and average the frequencies across loci. 1,000 replicate simulations were run, with results averaged across simulations to produce the 'simulation' line in Fig 1.

The prediction lines in Fig 1 were constructed by iterating Eqs. (5), (10), and (25), updating only the allele frequency each generation (i.e., holding $V_P$ constant, even though it is slowly decreasing according to the equations themselves).

**Methods for Fig 2.** The specification of the simulations in Fig 2 is the same as that in Fig 1, except that the loci now lie along sex-specific linkage maps. For humans, we used the male and female maps generated by Kong et al. [56], while for *Drosophila melanogaster*, we used the female map generated by Comeron et al. [57] (there is no crossing over in male *Drosophila*). In each case, we apportioned the $L = 1,000$ loci to chromosomes proportional to their physical (bp) lengths (as reported in build 38 of the human reference genome, available at https://www.ncbi.nlm.nih.gov/datasets/genome/GCF_000001405.26/, and in release 6 of the *D. melanogaster* reference genome, available at https://www.ncbi.nlm.nih.gov/datasets/genome/GCA_029775095.1/). We ignored the sex chromosomes and the fourth 'dot' chromosome of *D. melanogaster*. For each chromosome, we spaced the loci apportioned to it uniformly along its sex-averaged genetic (cM) length. This was done to avoid the occurrence of adjacent loci with a zero recombination fraction, since in such cases the harmonic mean recombination fraction among all locus pairs would be undefined. As is usual in SLiM, there is no

crossover interference; recombination fractions $r$ between adjacent loci were therefore calculated from the Morgan map distances between these loci $d$ via Haldane's map function, $r(d) = (1 - e^{-2d})/2$.

To plot the prediction line for Eq. (34) in Fig 2 requires calculating the harmonic mean sex-averaged recombination fraction for each locus, $\bar{r}_h^{(l)}$, and the overall harmonic mean sex-averaged recombination fraction, $\bar{r}_h$. For every pair of loci on the same chromosome, we calculated the recombination fraction from the map distance using Haldane's map function, as specified above; for pairs of loci on different chromosomes, the recombination fraction is $1/2$. The relevant averages of these recombination fractions were then taken. We calculated values of $\bar{r}_h$ for humans and *D. melanogaster* of 0.464 and 0.0744 respectively. Notice that, in averaging Eq. (34) across the $L = 1,000$ loci, we must first calculate $\left[1 - (d/V_g)(\bar{r}_h/\bar{r}_h^{(l)})\right]^2$ for each locus $l$ separately, and then average this quantity across loci.

**Methods for Fig 3**. The 'simulation' points in Fig 3 derive from the same simulations as in Fig 2 under the *D. melanogaster* linkage map. For each locus, the minor-allele frequency was measured after 250 generations, having started at frequency 0.2. We also calculated each locus $l$'s harmonic mean recombination fraction with all other loci, $\bar{r}_h^{(l)}$ —among the 1,000 loci, the minimum and maximum values were 0.063 and 0.133.

Because the dynamics at individual loci are noisy owing to drift, we binned loci according to their locus-specific harmonic mean recombination rates, into bins 0.05–0.07 (486 loci), 0.07–0.09 (388 loci), 0.09–0.11 (118 loci), 0.11–0.13 (6 loci), and 0.13–0.15 (2 loci). The simulation points in Fig 3 are the average minor-allele frequency in each bin after 250 generations, plotted against the (arithmetic) average value of $\bar{r}_h^{(l)}$ in each bin.

For the predictions in Fig 3 that take into account non-equilibrium values of LD after the onset of stabilizing selection (purple dots), note that Eq. (35) defines the sequence across generations $t$ of expected LD values between loci $l$ and $l'$, starting with zero LD in generation 0 ($D_{ll'}^{(0)} = 0$):

$$\mathbb{E}\left[D_{ll'}^{(t)}\right] = -\frac{d\bar{r}_h}{2\alpha^2 L(L-1)} \sum_{i=0}^{t-1} (1 - r_{ll'})^i = \frac{d\bar{r}_h}{2\alpha^2 L(L-1)} \cdot \frac{1 - (1 - r_{ll'})^t}{r_{ll'}}.$$

(42)

In each generation $t$, for each locus $l$, we calculate its expected LD values with every other locus $l'$ according to Eq. (42); we then sum these values and substitute the sums into Eqs. (15) and (16) to find the mean trait values experienced by the alleles at locus $l$. These define the effective effect size at locus $l$ in generation $t$,

$$\alpha_{\text{eff},l}^{(t)} = \alpha \left(1 + \frac{1}{p_l^{(t)}\left(1 - p_l^{(t)}\right)} \sum_{l' \neq l} \mathbb{E}\left[D_{ll'}^{(t)}\right]\right),$$

(43)

which then defines the expected change in allele frequency at locus $l$ from generation $t$ to $t + 1$:

$$\mathbb{E}\left[p_l^{(t+1)} - p_l^{(t)}\right] = p_l^{(t)}\left(1 - p_l^{(t)}\right)\left(p_l^{(t)} - 1/2\right)\frac{\left(\alpha_{\text{eff},l}^{(t)}\right)^2}{V_S + V_P}.$$

(44)

We calculated the expected allele frequency after 250 generations for each locus, and averaged these predictions within each bin to obtain the purple points plotted in Fig 3.

**Methods for S1, S2, S5, and S6 Figs**. In S1, S2, S5, and S6 Figs, minor-allele frequencies and effect sizes vary across loci. Initial minor-allele frequencies were chosen, independently for each locus, from a uniform distribution on [0.1, 0.3], and effect sizes were chosen, independently for each locus and independent of minor-allele frequencies, from a normal distribution with mean 0 and variance 1. Since effect sizes can be negative or positive here, we assigned the chosen effect size to the minor allele at each locus. The configuration results in an expected initial genetic variance of

$V_G$ = 313.3, and so we chose $V_S = 5V_G$ = 1566.7. In each replicate simulation, for the choice of minor-allele effect sizes and initial frequencies in that replicate, we calculated the initial mean trait value (which on average across replicates is zero) and defined this to be the optimal trait value in the replicate simulation; this procedure ensures that the population begins each replicate centered on its optimal trait value, consistent with our calculations above.

**Methods for S3 and S4 Figs.** The simulations used in S3 and S4 Figs are the same as those in Figs 1 and 2.

## Supporting information

**S1 Text. Supplementary information.**
(PDF)

**S1 Fig. Separation of timescales over which stabilizing selection reduces genetic variance by generating linkage disequilibrium (Bulmer effect; difference between genic and genetic variance) and selecting against minor alleles at polymorphic loci affecting the trait (reduction in the genic variance).** Here, loci are distributed along the human linkage map. Initial minor-allele frequencies were chosen from a uniform distribution on [0.1, 0.3] and effect sizes were chosen, independently, from a normal distribution with mean 0 and variance 1. The strength of selection $V_S$ was chosen such that $V_S/V_G$ = 5 initially. Trajectories are averaged across 1,000 replicate simulations.
(EPS)

**S2 Fig. Predictions of average allele-frequency trajectories, versus simulated trajectories, when effect sizes and initial minor-allele frequencies are allowed to differ, and all loci are unlinked.** Initial minor-allele frequencies were chosen from a uniform distribution on [0.1, 0.3] and effect sizes were chosen, independently, from a normal distribution with mean 0 and variance 1. The strength of selection $V_S$ was chosen such that $V_S/V_G$ = 5 initially. In each replicate simulation, the optimal trait value was taken to be the initial mean trait value in the population, given the chosen set of minor-allele effect sizes and initial frequencies. Otherwise, simulation details are as for Main Text Fig 1.
(EPS)

**S3 Fig. Average change in minor-allele frequency at polymorphic loci affecting a trait under weak stabilizing selection, of a strength of the order of that acting on human height.** Simulation details are as in Main Text Figs 1 and 2, but the strength of selection $V_S$ was chosen such that $V_S/V_G$ = 25 initially, and the simulation trajectories were averaged over 2,000 replicate trials rather than 1,000 owing to the larger effect of genetic drift relative to selection in this case. Again, Eqs. (25) and (34), which take into account the LD generated by stabilizing selection, are seen to provide a closer approximation to the observed trajectories than Eqs. (5) and (10), which do not.
(EPS)

**S4 Fig. Predicted changes in the genic and genetic variance, versus those observed in simulations.** For the genic variance predictions, we substituted Main Text Eqs. (5), (10), (25), or (34) into S1 Text Eq. (S5). Under this procedure, Eq. (25) (all loci unlinked; left panel) and Eq. (34) (loci distributed along the human and *D. melanogaster* genetic maps; right two panels), which take into account the LD generated by stabilizing selection, are seen to predict the change in the genic variance substantially better than Eqs. (5)and (10), which do not. Predictions of the change in the genetic variance are based on the predictions of the change in the genic variance, together with predictions of the change of the aggregate LD over time from Eq. (36) (see S1 Text Section S2). Since the predictions of the genetic variance necessarily involve LD, we only show predictions based on Eqs. (25)and (34), which are seen to be highly accurate. Simulation details are as in Main Text Figs 1 and 2.
(EPS)

**S5 Fig. Predictions of average allele-frequency trajectories, versus simulated trajectories, when effect sizes and initial minor-allele frequencies are allowed to differ, and loci are distributed along the autosomal linkage map of *Drosophila melanogaster* in the same way as in Main Text Fig 2B.** Initial minor-allele frequencies were chosen from a uniform distribution on [0.1, 0.3] and effect sizes were chosen, independently, from a normal distribution with mean 0 and variance 1. The strength of selection $V_S$ was chosen such that $V_S/V_G = 5$ initially. In each replicate simulation, the optimal trait value was taken to be the initial mean trait value in the population, given the chosen set of minor-allele effect sizes and initial frequencies. Otherwise, simulation details are as for Main Text Fig 1.
(EPS)

**S6 Fig. Predicted versus simulated minor-allele frequencies after 500 generations of selection when allelic effect sizes vary across loci, plotted as a function of the per-locus squared effect sizes.** Loci are distributed along the human linkage map in the same way as in Fig 3A. Initial minor-allele frequencies were chosen from a uniform distribution on [0.1, 0.3] and effect sizes were chosen, independently, from a normal distribution with mean 0 and variance 1. The strength of selection $V_S$ was chosen such that $V_S/V_G = 5$ initially. In each replicate simulation, the optimal trait value was taken to be the initial mean trait value in the population, given the chosen set of minor-allele effect sizes and initial frequencies. Otherwise, simulation details are as for Main Text Fig 1.
(EPS)

## Acknowledgments

We are grateful to Jeremy Berg, Graham Coop, Andy Dahl, Serena Debesai, Marida Ianni-Ravn, Xinyi Li, Pavitra Muralidhar, John Novembre, and Yuval Simons for helpful discussions.

## Author contributions

**Conceptualization:** Carl Veller.

**Formal analysis:** Sherif Negm, Carl Veller.

**Writing – original draft:** Sherif Negm, Carl Veller.

**Writing – review & editing:** Sherif Negm, Carl Veller.

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
