## [Decision Letter · Decision Letter 0]

10 Nov 2025

PGENETICS-D-25-01059

The effect of long-range linkage disequilibrium on allele-frequency dynamics under stabilizing selection

PLOS Genetics

Dear Dr. Veller,

Thank you for submitting your manuscript to PLOS Genetics. After careful consideration, we feel that it has merit but does not fully meet PLOS Genetics's publication criteria as it currently stands. Therefore, we invite you to submit a revised version of the manuscript that addresses the points raised during the review process.

Please submit your revised manuscript within by Dec 10 2025 11:59PM. If you will need more time than this to complete your revisions, please reply to this message or contact the journal office at plosgenetics@plos.org. Please include the following items when submitting your revised manuscript:

We look forward to receiving your revised manuscript.

Kind regards,

Kirk E Lohmueller

Academic Editor

PLOS Genetics

Justin Fay

Section Editor

PLOS Genetics

Aimée Dudley

Editor-in-Chief

PLOS Genetics

Anne Goriely

Editor-in-Chief

PLOS Genetics

**Additional Editor Comments :**

Thank you for submitting your manuscript to PLoS Genetics. It has been evaluated by three reviewers whose comments are below. All reviewers found a number of positive aspects of your manuscript and that it could be suitable for PLoS Genetics. I concur with this assessment.

Reviewers 1 and 2 have fairly minor comments that should be easily addressable.

Reviewer 3 has the most extensive comments. In particular, please pay special attention to their comments about whether the conditions shown in Figs 1 and 2 are likely to actually occur in real populations. While Reviewer 3 suggests that space could be saved by not re-deriving many of the classic results, I think there is value clearly presenting these results, as they are quite technical (as noted by Reviewer 1).

Please revise your manuscript to address all of the reviewer comments, paying special attention to the ones that I highlighted above.

**Journal Requirements:**

At this stage, the following Authors/Authors require contributions: Sherif Negm, and Carl Veller. Please ensure that the full contributions of each author are acknowledged in the "Add/Edit/Remove Authors" section of our submission form.

The list of CRediT author contributions may be found here: https://journals.plos.org/plosgenetics/s/authorship#loc-author-contributions

https://journals.plos.org/plosgenetics/s/submission-guidelines#loc-parts-of-a-submission

5) We notice that your supplementary Figures, and information are included in the manuscript file. Please remove them and upload them with the file type 'Supporting Information'. Please ensure that each Supporting Information file has a legend listed in the manuscript after the references list.

6) Please ensure that the funders and grant numbers match between the Financial Disclosure field and the Funding Information tab in your submission form. Note that the funders must be provided in the same order in both places as well.

Please amend your detailed Financial Disclosure statement. This is published with the article. It must therefore be completed in full sentences and contain the exact wording you wish to be published.

1) Please clarify all sources of financial support for your study. List the grants, grant numbers, and organizations that funded your study, including funding received from your institution. Please note that suppliers of material support, including research materials, should be recognized in the Acknowledgements section rather than in the Financial Disclosure

2) State the initials, alongside each funding source, of each author to receive each grant. For example: "This work was supported by the National Institutes of Health (####### to AM; ###### to CJ) and the National Science Foundation (###### to AM)."

3) State what role the funders took in the study. If the funders had no role in your study, please state: "The funders had no role in study design, data collection and analysis, decision to publish, or preparation of the manuscript."

4) If any authors received a salary from any of your funders, please state which authors and which funders.

**Reviewers' comments:**

Reviewer's Responses to Questions

**Comments to the Authors:**

**Please note that one review is uploaded as an attachment.**

Reviewer #1: In "The effect of long-range linkage disequilibrium on allele-frequency dynamics under stabilizing selection," Negm and Veller present a theoretical advancement for predicting allele frequency changes when those alleles contribute to stabilizing selection. Stabilizing selection is now understood to be important in explaining quantitative trait evolution and interpreting GWAS, and this work is a nice contribution towards gaining a population genetic perspective of common-place selection on complex traits. I believe it is sound and timely work, and I would be happy to see it published more-or-less as is.

I found no issues with the methods or theory, and the exposition is clear. I especially liked the discussion, which places their results in the broader context of other aspects affecting quantitative trait architectures, namely assortative mating and pleiotropy. While the paper lacks a direct application to empirical data, I don't believe it is warranted here as the theoretical advances help us in interpreting existing results in the literature and would be beyond the scope of this work.

The paper is somewhat technical, drawing from existing theory stretching back to the 1970s (Bulmer), but I think it is presented in a way that is not too unapproachable. For readers unfamiliar with this particular literature (which is deep and technical, as the authors point out), the paper could present a challenge. It's unavoidable that notation becomes a bit of a burden, so in editing, I would just suggest focusing on clarifying notation multiple times throughout, especially wrt Bulmer's original notation (d, X, etc). I think the authors already do a good job of this, really.

My only other comment is about the choice of parameters in the simulations. The authors set VS/VP=5, which I would have thought implies a whole lot of phenotypic variation relative to the strength of selection. If we take VG=4UVS (approx), this might imply a very high mutation rate and thus polygenicity. For height, as discussed on line 532, VS/VP~20-30, and we think of height already as highly polygenic. A value of 5 might be unreasonably small, at least for interpreting human quantitative genetics results. My guess is that the authors chose such a value to amplify the effects they are modeling, which is understandable for a theory paper and demonstrating the accuracy of their results. But I could easily be wrong about this - justifying this choice of parameter values would be helpful.

Reviewer #2: The manuscript by Negm and Veller uses analytical theory and simulations to study the impact of linkage disequilibrium on allele frequency changes under a model of stabilizing selection. They show that previous expressions under-estimate the change in allele frequencies. They also re-derive some well known results in an effort to reintroduce them to (human) geneticists. I found the manuscript to be well-written and the results interesting and I believe it is a good candidate for publication in PLOS Genetics. I have some suggestions below, but I do not think that they are necessary for publication.

- Do the more accurate allele frequency dynamics do a better job of also predicting the change in additive genetic variance? If so, I think this would be sufficiently interesting to add to the main text.

- I am curious about the impact of demographic history on these results (particularly: bottlenecks, population growth, population structure+migration). I understand that carrying out simulations to explore this would be a large undertaking. Instead, I wondered if the authors could add a discussion of the expected impact of demography on their results. This would help connect the discussion on polygenic risk score portability.

- In the discussion of allelic turnover, I thought the authors may want to cite a couple of additional papers that touch on this point: PMID: 31201529, PMID: 27197206, PMID: 33691092.

- In addition, in the discussion of signed LD, a recent study has looked at this empirically in the UK Biobank: PMID: 38106023.

- The discussion of pleiotropy is very interesting and I wonder if the model gives the authors an opportunity to set some bounds based on LD about the total number of indepdent traits that could be under selection?

Reviewer #3: Review is uploaded.

**Have all data underlying the figures and results presented in the manuscript been provided?**

Reviewer #1: Yes

Reviewer #2: Yes

Reviewer #3: Yes

PLOS authors have the option to publish the peer review history of their article (what does this mean? ). If published, this will include your full peer review and any attached files.

**Do you want your identity to be public for this peer review?** For information about this choice, including consent withdrawal, please see our Privacy Policy .

Reviewer #1: No

Reviewer #2: No

Reviewer #3: No

**Figure resubmission:**
---

## [Editor Report · Decision Letter 1]

17 Jan 2026

Dear Dr Veller,

We are pleased to inform you that your manuscript entitled "The effect of long-range linkage disequilibrium on allele-frequency dynamics under stabilizing selection" has been editorially accepted for publication in PLOS Genetics. Congratulations!

Yours sincerely,

Kirk E Lohmueller

Academic Editor

PLOS Genetics

Justin Fay

Section Editor

PLOS Genetics

Aimée Dudley

Editor-in-Chief

PLOS Genetics

Anne Goriely

Editor-in-Chief

PLOS Genetics

BlueSky: @plos.bsky.social

Comments from the reviewers (if applicable):

**Data Deposition**

http://datadryad.org/submit?journalID=pgenetics&manu=PGENETICS-D-25-01059R1

**Press Queries**

---

## [Editor Report · Acceptance letter]

PGENETICS-D-25-01059R1

The effect of long-range linkage disequilibrium on allele-frequency dynamics under stabilizing selection

Dear Dr Veller,

We are pleased to inform you that your manuscript entitled "The effect of long-range linkage disequilibrium on allele-frequency dynamics under stabilizing selection" has been formally accepted for publication in PLOS Genetics! Your manuscript is now with our production department and you will be notified of the publication date in due course.

With kind regards,

Anita Estes

PLOS Genetics

On behalf of:
